# Clinical Manifestations and Changes of Haematological Markers among Active People Living in Polluted City: The Case of Douala, Cameroon

**DOI:** 10.3390/ijerph18020665

**Published:** 2021-01-14

**Authors:** Tiekwe Joseph Eloge, Ongbayokolak Nadine, Dabou Solange, Phélix Bruno Telefo, Isabella Annesi-Maesano

**Affiliations:** 1Department of Biochemistry, University of Dschang, Dschang, P.O Box 67 Dschang, Cameroon; tiekweeloge@gmail.com (T.J.E.); nongbayokolak@yahoo.fr (O.N.); solangedabs@gmail.com (D.S.); 2Department of Epidemiology of Allergic and Respiratory Diseases (EPAR), IPLESP INSERM et Sorbonne Université, 15-21 Rue de l’École de Médecine, 75006 Paris, France; isabella.annesi-maesano@inserm.fr

**Keywords:** urban air pollution, clinical manifestations, haematological markers, active people

## Abstract

Urban air pollution, despite its dangerous health impact, is poorly studied in sub-Saharan Africa (sSA). Epidemiological data on this silent killer are almost non-existent for cities of Cameroon, which seems to be one of the sSA countries where populations are highly exposed to air pollutants. *Objective*: The present study was conducted in Douala city, and aimed at determining the association of urban air quality degradation with respiratory and systemic health in active populations exposed to air pollutants on a daily basis. *Methods:* A cross-sectional study was conducted from 2017 to 2019 in 1182 active people consisting of motorbikes drivers (MD), outdoor urban workers (UW), and fuel station sellers (FSS). A standardized questionnaire was used to document participants’ data. One hundred and twenty-six (126) motorbike drivers were selected to evaluate the relationship between haematological (white blood cells, platelets) and inflammatory (C-reactive protein—CRP) biomarkers, and air pollution; compared with those of a sixty-five (65) motorbike drivers’ control group enrolled in Dschang, another town situated at about 216.3 km from Douala. *Results*: Among those recruited in urban Douala, some respiratory disorders such as running nostrils, colds, common fever, sore throats, dry cough, wheezing, chest pain, shortness of breath and systemic symptoms such as headaches, eye irritation, conjunctivitis, watery eyes and general tiredness were very common among MD, UW, and FSS. Regarding biological data, blood monocytes, lymphocytes and CRP were found to be significantly increased among selected MD in Douala, compared to control groups in Dschang. Conversely, a more significant decrease in blood neutrophil level was observed among MD in Douala than control groups in Dschang. These changes of haematological markers were significantly associated with place of residence, site of activity, and daily duration. *Conclusion*: Our results suggest the risk of suffering from respiratory impairments and systemic symptoms with exposure to urban air pollution among active people working near highways in Douala.

## 1. Introduction

Air pollution (AP) has become a real problem of public health, with several adverse effects in exposed people. Some of air pollutants result from human activity, mainly industrial and urban traffic activities. They are grouped as primary or secondary pollutants. The first group is directly produced and includes mainly particulate matter (PM) such as PM_10_ and PM_2.5_, carbon monoxide (CO), nitrogen oxides (NO_x_), or sulfur dioxide (SO_2_), volatile organic compounds (VOCs), and polycyclic aromatic hydrocarbon (PAH). The second group stems from the interaction between primary pollutants and includes compounds such as ozone (O_3_) resulting from the interaction between NO_x_ and VOCs [1]. According to the Lancet Commission on pollution and health, about 9 million people die every year, attributable to degraded environmental conditions, of which about half to ambient (outdoor) AP. In Europe, although surveillance systems are present in most cities, the annual excess mortality rate from ambient air pollution is 790,000 [2]. About 40% and 80% are due to cardiovascular events, which dominate health outcomes [2]. In most of sub-Saharan African (sSA) cities, populations suffer from the highest burden of air quality-deteriorated related disease and premature death [3].

Many recent studies showed an increased risk of developing respiratory and cardiovascular diseases (e.g., upon long term exposures to urban air [4,5]. The effects are more pronounced among active people, individuals with anterior pulmonary diseases, children, and aging people [4]. Other studies showed a relation between urban AP and changes of haematological parameters in healthy individuals; notably increase in white blood cell and platelet numbers as well as pro-inflammatory state related to cardiometabolic risk factors [6] with pro-inflammatory state related to cardiometabolic risk factors. It has also been revealed a decreasing of total white blood cell, lymphocyte and eosinophil counts among exposed motorbike drivers (MD) compared to unexposed controls in Benin [7]. Furthermore, one study pointed out not only, an immediate decrease in polymorphonuclear leukocytes in response to an increase of most gaseous and particulate pollutants, but also an increase in lymphocytes and monocytes in association with all gaseous pollutants, ultrafine particles, and NO_x_, respectively, particularly in people with chronic pulmonary diseases [8]. On the other hand, inflammatory response is implicated as a biological mechanism that links PM with health effects [9]. C-reactive protein (CRP), which is an important acute phase reactant with profound proinflammatory properties, is used clinically as an indicator of the presence and intensity of inflammation studies, suggesting that CRP levels increase in response to PM exposure [10]. Elevated CRP levels were consistently found among children, and healthy adults, although requiring higher peak levels of PM exposure [11].

Urban air pollutant-induced health effects are sometimes the cause of high mortality and morbidity in most low-income sSA countries [3]. There is evidence that the health effects of AP in urban settings remain underestimated, because of lack of air quality monitoring in most metropolitan sSA cities, and the paucity of epidemiological studies on AP [3]. In Cameroon, to the best of our knowledge, there are no epidemiological data concerning health effects of urban AP.

Douala, the country economic capital, is confronted with AP-related high motor traffic and industrial waste. There has been rapid urban demographic growth with an explosion of exodus of people from rural setting to urban areas for a better quality of life. Nearly 80% of the industrial activities in this town have a major seaport supplying almost all the Central Africa countries; this increases the density of road traffic, and AP as well. The intense urban traffic is often the result of thousands of inappropriate vehicles and motorbikes more than 15 years old, often fueled by poor quality fuel. Those contribute to change air quality of this city with the risk of pulmonary and cardiovascular diseases for the actively exposed people. Our study was carried out from 2017 to 2019 at Douala, aimed at assessing distribution of clinical manifestations and biological systemic and inflammatory markers changes among daily active people in Cameroon.

## 2. Materials and Methods

A cross-sectional study and a nested study were conducted to determine the relationship between exposure to air pollutants and health outcomes among active individuals.

### 2.1. Study Site

Douala, the economic capital of Cameroon with about 2.8 million inhabitants is the main and the largest metropolitan urban city of the country, with an area about 210 square kilometers [12]. On the other hand, Dschang is located in the West of country, and is about 216.3 km from Douala. It is a student town in which industrial activity is almost inexistent [13] and where most roads are stony. In this context, Dschang was considered as the control area, to evaluate the effect of AP on health status. An estimated of 70,000 inhabitants live in this town. The main activity in Dschang is agriculture and breeding.

### 2.2. Target Population and Sampling

Participants recruited in this study were apparently healthy people aged from 21 to 40 years living in Douala and Dschang. Three categories of population were targeted for the study: motorbike drivers (MD), outdoor urban workers (UW), and fuel station sellers (FSS).

They were recruited at the level of crossroads and nearby industrial plants, which are their daily places of activity from 2017 to 2019, and replied to an interviewer-administered questionnaire. In the nested study, MD recruited in Douala and in Dschang, were selected following the number of years of activity and the place of residence. There were invited to health centers for blood sampling and analysis of haematological (red and white blood cells, platelets), and inflammatory biomarkers. Blood sampling and anthropometric parameters (height, weight, blood pressure) were performed by hospital nurses. The body mass index (BMI) of each participant was computed using height and weight measurements.

### 2.3. Eligibity Criteria

Any apparently healthy adult individual and permanently living in the study town for at least five years were included in the study. Pregnant women, lactating women, individuals under treatment against tuberculosis, cancer respiratory and heart diseases, hepatitis, lung or kidney ailments; and individuals using partial or permanent oxygen supply were excluded from the study.

In the nested study, motorbike drivers selected for blood sampling were those who do not smoke, had about 10 years of service, and 14 h of activity per day. The control group comprised motorbike drivers living in Dschang, with almost the same characteristics as those of Douala.

### 2.4. Questionnaire

The questionnaire of the study was based on respiratory questionnaires of the British Council for Medical Research (BMRC), the American Thoracic Society (ATS) and the American National Institute of Heart and Lung Division adult lung disease (ATS-DLD-78-C), respectively. It was divided into four parts:-Socio-demographic characteristics of participants such as gender, age, and level of education;-Others factors such as consumption of tobacco and alcohol, use of coal, firewood, and domestic gas;-Exposure assessment: time and places of exposure (in years, per daily hour) to different factors such as consumption habits (alcohol, tobacco), and cooking method, use of coal, firewood, and domestic gas;-Health outcomes and biological parameters.

### 2.5. Symptoms

Sixteen (16) symptoms were identified and divided into three categories:-Symptoms or diseases related to upper airways such as sinusitis, pungent nostrils, cold, current fever and sore throat;-Symptoms or diseases related to lower respiratory system damage such as dry cough, wheezing, thoracic painful, breathlessness;-Systemic symptoms or diseases such as headache, dizziness, eyes pain, conjunctivitis, lacrimation, general tiredness, and nausea.

The questionnaire was prepared in French and English, and administered to participants in clearly understanding language. The questionnaires were completed for illiterate and poorly educated participants.

### 2.6. Anthropometric and Biological Parameters

Biological parameters taken into account in this study were haematological markers such as total red blood cell (RBC) count (×10^6^/μL), haemoglobin (Hb) content (g/dL), total white blood cell (WBC) count (×10^3^/μL) and platelet count (×10^3^/μL); and CRP.

Total RBC count, Hb content, total WBC count and platelet count were assessed using Urit-2900 Plus hematology analyzer (Urit Medical Electronics, Guangxi, China). Measurement of CRP was carried out by a latex agglutination slide test. The principle of this test was based on mixing serum and CRP latex reagent, allowing them to react. A visible agglutination is observed for a CRP concentration more than 0.6 mg/dL.

### 2.7. Ethical Considerations

This study was carried out in conformance with the guidelines for human experimental models in clinical research as stated by the Cameroon Ministry of Public Health and the Helsinki declaration. An ethical clearance was obtained from the institutional review board (IRB) of the University of Douala under the registering number N°1751 CEI-Udo/04/2019/T. The aim and objectives of the study were explained to them in the language they understood best (French or English), and their questions were answered. Only individuals who signed an informed consent form for their participation were enrolled. Participation in the study was strictly voluntary and patients were free to decline answering any question or totally withdraw if they so wished at any time.

### 2.8. Statistical Analyses

The data were keyed, classified, checked for consistency and analyzed using statistical package for social science (SPSS) version 20.0 (IBM, Armonk, NY, USA). Frequencies and Mean ± standard deviation (SD) were computed where appropriate. Pearson’s independent Chi-square and Fisher’s exact tests were used for comparing proportions. Significant differences between mean values of study and control group were statistically analysed using Student’s *t*-test. Bivariate correlation using Pearson’s test was used for evaluating association between sites and time of exposure, and changes of haematological parameters. Then, linear regression and multinominal analysis were used for assessing associations between sites, time of exposure (in year and hour), after adjustment for age, level of education, BMI, smoking and use of domestic gas, firewood and coal, and each category of biological parameter. The significance was set at *p*-value < 0.05 and <0.001.

## 3. Results

### 3.1. Cross-Sectional Study

A total of 1182 people were recruited, including 895 motorbike drivers (MD), 53 fuel station sellers (FSS), 120 outdoor urban workers (UW), and 114 workers in closed places (WCP) such as trade shops, and small-scale dealers.

The 1182 participants recruited in the setting of this study consisted in the majority of males (91.37%), and with a mean age of 31.49 ± 8.88 years (*p*-value < 0.001). Almost all participants completed secondary school studies (61.42%), (Table 1, *p*-value < 0.001). Almost 80% of these participants daily spent 7 to 14 h per day at their place of activity (Table 2, *p*-value = 0.0527). Most of participants declared never having smoked tobacco notably among motorbike drivers (69.94%), ODW (69.29%), FSS (84.90%), and WCP (87.50%) (*p*-value < 0.001) (Table 2). Almost 72.58% of them were regular consumers of alcohol. Furthermore, most of the participants used domestic gas (90.35%) as a cooking method comparatively to other methods such as firewood (26.4%) and, coal (31.13%). This observation is significantly different among all categories of participants (*p*-value < 0.001) (Table 2).

Upper respiratory problems such as running nostrils, colds, common fever and sore throat were commoner among motorbike drivers, than people practicing in urban areas and fuel station sellers (*p*-value < 0.001). The strong distribution of dry cough, wheezing, chest pain, and shortness of breath was observed among motorbike drivers. In people who are active near roads, dry coughing and wheezing were more common. Fuel station seller presented a higher frequency of dry coughing and shortness of breath (*p*-value < 0.001). Systemic symptoms as headache, eye irritation, conjunctivitis, watery eyes and general tiredness were widely distributed among motorbike drivers and people who work daily near highways (*p*-value < 0.001) (Table 3).

### 3.2. Nested Study

The distribution of MD and group control in this section of the paper is presented in Table 4. The proportions of the different AP-polluted symptoms were significantly higher in MD from Douala compared to their counterparts from Dschang. Regarding symptoms of upper respiratory airways including running nose (25.61% in Douala vs. 25% in Dschang, *p*-value = 0.015), cold (44.62% vs. 42.50%, *p*-value < 0.0001), and sore throat (12.80% vs. 5.0%, *p*-value = 0.0002) (Table 4). The same pattern of significantly increased proportion in MD from Douala, was found for some symptoms of lower respiratory airways (dry cough, chest discomfort, and breathless), and systemic symptoms (headache, eye irritation, conjunctivitis, watering, and general tiredness) (see details in Table 4). Discomfort (*p*-value = 0.020), and breathlessness (*p*-value < 0.001), were significantly more distributed among motorbike drivers recruited in Douala than those of Dschang. On the others hand, symptoms associated with general discomfort such as headache (*p*-value < 0.001), eyes irritation (*p*-value < 0.001), conjunctivitis (*p*-value = 0.021), general tiredness (*p*-value < 0.001), watering (*p*-value < 0.001) were significantly more distributed among motorbike drivers in Douala and nausea (*p* = 0.051) appears significantly more distributed among those of Dschang.

Mean and standard deviation of arterial tension, haematological parameters and C-reactive protein are presented in Table 5.

The motorbike drivers (MD) from Douala and those of the control group in Dschang were all young with a mean age of 29.87 ± 5.47 and 30.00 ± 5.17 years, respectively. Their anthropometric parameters revealed a significantly higher proportion of overweight people in MD from Douala compared to those from Dschang. Their systolic blood pressure (SBP) and diastolic blood pressure (DBP) were both 138.49 ± 16.57 mmHg and 87.72 ± 14.46 mmHg, respectively.

Blood analysis showed a significant increase of monocytes (15.56 ± 6.24—range 0.0–10), and a significant decrease of neutrophils 34.50 ± 11.56 (range 42.0–85.0) in MD from Douala compared to control group. The levels of RBC (4.98 ± 0.53—range 4.5–5.9 × 10^12^ Cell/L), Hb (14.7 ± 1.55—range 13.0–17.0 g/dL) and platelets (234.82 ± 39.55—range 150–500 × 10^9^ Cell/L) remained unchanged. Concentrations were in the reference values. However, the blood of participants had a good colour, depending on value of the Hct which was 43.99 ± 4.76% (range 39–54). Lymphocytes and CRP were increased among motorbike drivers in Douala and the control groups in Dschang.

Table 6 shows a significant influence of place of residence, site and daily duration (per hour) of activity on changes in neutrophils, lymphocytes and monocytes. The duration of activity (per year), was significantly and positively associated with variation in blood CRP (*r* = 0.311, *p* = 0.0001).

A significant negative association was observed between place of residence (R = −0.194, *p*-value = 0.030), site of activity (R = −0.194, *p* = 0.030) and monocytes. The same line was observed between daily duration (R = −0.28, *p*-value = 0.002) and the variation of lymphocytes. On the other hand, a statistically significant and positive correlation was found between neutrophils count and site of activity (R= 0.21, *p*-value = 0.019), and daily duration (per hour) (R = 0.199, *p*-value = 0.026). A significant positive influence was observed of place of residence (R = 0.26, *p*-value = 0.004) and duration of activity (per year) (R = 0.31, *p*-value = 0.00001) on changes of lymphocytes and CRP, respectively.

Regarding linear regression and multinominal analysis in Table 7, age, place of residence, and duration of activity (per year) influence change of CRP, RBC, Hb, monocytes and lymphocytes of MBD in Douala, after adjustment for age, level of education, BMI, smoking, and use of domestic gas, firewood and coal.

The change of monocytes (*p*-value = 0.028) and lymphocytes (*p*-value = 0.023) was significantly more increased among the youngest MD. Place of residence, influence significantly increase of CRP (*p*-value = 0.011), RBC (*p*-value = 0.0001), Hb (*p*-value = 0.004), and monocytes (*p*-value = 0.0001). Furthermore, CRP (*p*-value = 0.0001) was also significantly increased with an increase of duration of activity (per year)

According to Table 8, Table 9 and Table 10, a significant association was observed between distribution of sinusitis (R = 0.332, *p*-value = 0.0001), cold (R = −0.315, *p*-value = 0.0001), dry cough (R = −0.183 *, *p* = 0.041), wheezing breath (R = 0.219, *p*-value = 0.014), and changes in CRP, neutrophils, and monocytes. The distribution of other symptoms was not significantly associated with the changes in haematological markers.

## 4. Discussion

This study carried out in Douala and a control group in Dschang, shows the role of air pollution exposure on the health of active people in the urban surroundings. Indeed, symptoms related to upper and lower airways affections appeared more distributed among motorbike drivers and outdoor urban workers in Douala. The same was observed with symptoms related to discomfort, mostly distributed among motorbike drivers of Douala. These results corroborate with those of epidemiological study of the Central Pollution Control Board (CPCB), carried out in 2012 among adults recruited in Delhi, India. In this study, symptoms appeared more pronounced in adults recruited near areas with high urban traffic [14]. This suggests the role that proximity with sources of pollution can play in the distribution of pathologies. Douala, like most metropoli in developing countries, is experiencing a deterioration of air quality. Recent studies study have has shown that some streets and crossroads of Douala have PM and gaseous pollutants concentrations about 4 to 8 times higher than World Health Organization (WHO) standards [15]. This could be mainly due not only to the high urban traffic with a large number of vehicles and motorbikrs over 15 years old, sometimes fuelled by poor quality fuels, but also by different industrial facilities, household pollution from biomass burning around the city, public embers in several places, and domestic cooking [15,16]. Douala is a town far from agricultural activities; therefore, agricultural emissions are almost non-existent, and cannot contribute to the effects on the health of the population. The different sources emit a set of long-term harmful air pollutants to exposed populations [14,15]. Several studies revealed that PM affect the upper part of the respiratory system, causing irritation of the trachea, sore throat, and flowing nostrils, among most exposed people [11,17]. These PM are harmful to health because they can block and inflame nasal and bronchial passages, thus causing a variety of respiratory-related conditions that lead to illness or death. This may explain the greater distribution of these symptoms among motorbike drivers and outdoor urban workers of this study, who have experienced a significantly longer duration of air exposure per year and hour. Until now, a strong relationship was established between concentration of air pollutants and the occurrence of health-related side effects. Smaller diameter PMs penetrate deep into the lower part of the respiratory system with consequences ranging from chronic obstructive pulmonary disease to reduced lung function [4,10,14,18]. Affections often manifest as dry cough, wheezing, chest pain, and shortness of breath in people who are exposed to poor air quality in the long term. Clinical manifestations vary from exposure time, pollutant concentration, smoking and history of lung disease [14,18]. However, the finest particles pass through the lining of the pulmonary alveoli, and end up in the bloodstream, affecting other internal organs such as the heart, liver, kidney and brain [14,18], with a strong mobilization of haematological markers such as polynuclear neutrophils, monocytes, lymphocytes, and platelets [14,19]. The results of biological parameters of this study, confirm this evidence.

Indeed, the blood proportion of neutrophils and monocytes were significantly decreased and increased respectively among recruited motorbike drivers, compared to the control group. The lymphocytes were also increased among the two groups. These results corroborate with those of Parinaz and al., 2011 [20] who showed the significant associations between exposure to air pollution and variations of haematological parameters, with an increase of white blood cells in a population-based sample of children and adolescents. It the importance of PM_10_ in the blood mobilization of white blood cell at alveolar level is revealed in this study. The previous study in Douala showed an increase of PM_10_ 8 times more than the WHO limit values at the level of Rond-point, Deido crossroads [15], one of main sites where the motorbike drivers were recruited. In the study of Irene et al., 2010, in patients with chronic pulmonary disease, not only is an immediate decrease of polymorphonuclear leukocytes observed in response to an increase of most gaseous and particulate pollutants, but also an increase of lymphocytes within 24 h in association with all gaseous pollutants but showed no effect in regard to particulate air pollution [8] As for the monocytes, an increase is revealed and associated with ultrafine particles, and nitrogen monoxide. Along the same lines, the study of Steenhof et al. 2014 revealed changes in total white blood cells counts between 2 and 18 h of air pollution exposure, number of neutrophils for 2 h of exposure and monocytes for 18 h of exposure, associated with PM characteristics [6]. It a decrease of lymphocytes was observed, with an increase of NO_2_ [6]. Our study showed the influence not only of place of residence, site of activity, but also duration of activity (per year) and daily duration (per hour) on changes of neutrophils, monocytes lymphocytes and C-reactive protein in selected motorbike drivers. In this study, motorcycle drivers had a daily exposure time of about seven hours and mostly took more than 10 years to operate. The results of Avogbe et al., 2011, show that exposed motorbike drivers had low total WBC, lymphocytes and eosinophils counts than controls, with variation of air benzene, and following an exposure more than five hours per day [7]. However, our study did not reveal any changes of red blood cells, haemoglobin, and haematocrit with air pollution exposure in motorbike drivers. These results are the same as those of Avogbe et al., 2011, which showed exposure to benzene do not modify the numbers of red blood cells among motorbike drivers. On the other hand, the change of C-reactive protein (CRP) among motorbikes drivers and controls groups reveals a systemic inflammatory response including stimulation of the bone marrow and progression of atherosclerosis with the risk of development of cardiovascular diseases. The finding are in line with several studies showing, thus, the implications of inflammatory response as a biologic mechanism that links PM air pollution with health effects [11,21]. Indeed, CRP an important acute-phase reactant with profound proinflammatory properties, is used clinically as an indicator of the presence and intensity of inflammation. In vitro and in vivo animal studies suggest that CRP levels increase in response to PM exposure, but there was no consistency in epidemiologic studies [9]. Some studies revealed that changes of CRP levels are observed among children, and healthy adults with higher peak levels of PM exposure [10,11,17,22]. That corroborates our study, which showed a significant increase of CRP levels among motorbike drivers and unexposed controls groups. The motorbike drivers were recruited in the sites high polluted in PM. Although control groups have been recruited from an area considered less polluted, it is still an area at risk for pathologies that could cause CRP levels to vary [10]. Therefore, it is necessary to reduce the confounding factors among these participants, to finally establish a close link between exposure to PM and the establishment of a systemic inflammation, characterized by the variation in the level of CRP. It becomes important to multiply epidemiological studies in this direction by increasing the number of participants, taking into account the characteristics of the participants and checking the confounding factors that could cause haematological and inflammatory parameters to vary.

Therefore, some biomarkers such as certain molecules secreted by the lungs (antioxidants present in the respiratory tract lining fluid, cell clara protein cc16), the most sensitive inflammatory molecules, and stress oxidative biomarkers must to be added in the diagnosis of disorders related to air pollution. That will permit to early determine the direct link between the degradation of air quality in an area and the health consequences of exposed populations, eliminating any other cause for these disorders.

## 5. Conclusions

This study aimed at depicting the relation between clinical manifestations and blood haematological and inflammatory markers with air pollution among active people in Douala, Cameroon. At least one of these manifestations was reported in the participants. Most of the symptoms occurred in participants near high polluted sites such as crossroads with high levels of urban traffic. This showed a strong presumption of a correlation between the degradation of air quality and the birth of several pathologies among these active people at Douala. The long-term exposure at the level of polluted sites poses a risk of developing lung diseases, discomfort and cardiovascular disorders, with the negative changes of some haematological and biochemical parameters among exposed people. The significant influence of place of residence was observed, and duration of activity (per year) on changes of C-reactive protein, red blood cell, hemoglobin, monocytes and lymphocytes of motorbike drivers in Douala, after adjustment for age, level of education, BMI, smoking and use of domestic gas, firewood and coal. However, our study has a certain number of weaknesses even though it has scientific relevance. First, this study was exclusively questionnaire-based. Thus, a proportion of participants would have given the wrong answers. Second, other causes could be associated with manifestations documented in the study. These are malaria, tuberculosis, viral and helminths infections which can be responsible for symptoms such as headache, tiredness and respiratory disorders. On the other hand, this pilot study is the first to address the problem of urban air pollution in a Cameroonian context. Thus, further studies of our research team on the topic will be carried out in order to provide baseline epidemiological data with an increase in the number of subjects. Strategies could be implemented to efficiently manage the problem of air pollution. There is a need to appraise the quality of air in Douala and Dschang along with educating the population about air pollution and preventive methods.

## Figures and Tables

**Table 1 ijerph-18-00665-t001:** Baseline characteristics of participants.

**Gender *n* (%)**		**MD *n* = 895**		**UW *n*** **= 114**	**FSS *n* = 53**	**WCP *n* =120**	**Total**	**χ^2^ (*p*-Value)**
		**M**	895 (100)	82 (71.92)	41 (77.35)	62 (51.66)	1080 (91.30)	
		**F**	0	32 (28.07)	12 (22.64)	58 (48.33)	102 (8.62)	
**Age groups (years) *n* (%)**						
(21–27)		232 (25.92)		49 (42.98)	23 (41.10)	76 (63.33)	380 (32.14)	50 (*p* < 0.001 **) df = 9
(28–33)		259 (29.00)		33 (28.90)	10 (17.90)	15 (12.50)	317 (26.81)	
(34–39)		227 (25.40)		10 (08.70)	18 (32.10)	9 (07.50)	264 (22.33)	
≥ 40		177 (19.80)		22 (19.30)	2 (3.60)	20 (16.67)	221 (18.69)	
**Total**		**895**		**114**	**53**	**120**	**1182**	
**Mean of age (±SD)**		34.01 (±8.97)		30.63 (±8.77)	30.43 (±6.544)	30.92 (± 11.26)	31.49 (±8.88)	
**Level of education *n* (%)**						
None		34 (03.70)		8 (07.00)	0	3 (02.50)	45 (3.80)	82.73 (*p* < 0.001 **) Df = 9
Primary		190 (21.30)		37 (32.50)	0	17 (14.17)	244 (20.64)	
Secondary		576 (64.40)		53 (46.50)	35 (62.50)	62 (51.67)	726 (61.42)	
University		95 (10.60)		16 (14.00)	18 (32.10)	38 (31.66)	167 (14.12)	
**Total**		**895**		**114**	**53**	**120**	**1182**	
		**MD *n* = 895**		**UW *n* = 114**	**FSS *n* = 53**	**WCP *n* = 120**	**Total *n* = 1182**	**χ2 (*p*-Value)**
**Tobacco smoking**	Current smoker	211 (23.57)		35 (30.70)	6 (11.32)	11 (9.16)	263 (22.25)	512.108
	EX-smoker	58 (6.48)		0 (0.00)	2 (3.77)	4 (3.33)	64 (5.41)	(*p* < 0.00 1 **)
	Never smoker	626 (69.94)		79 (69.29)	45 (84.90)	105 (87.50)	855 (72.33)	df = 6
**Habit of Alcohol**	Regular use	734 (82.01)		5 (4.38)	32 (60.37)	87 (72.50)	858 (72.58)	681.875 (*p* < 0.001 **) df = 6
	Temporary	18 (2.01)		88 (77.19)	21 (39.62)	3 (2.50)	130 (10.99)	
	Never	143 (15.97)		21 (18.42)	0 (0.00)	30 (25.00)	194 (16.41)	
**Domestic gas**	yes	793 (88.60)		114 (100)	53 (100)	106 (88.33)	1068 (90.35)	21.229 (*p* < 0.001 **) df = 3
	No	102 (11.39)		0 (0.00)	0 (0.00)	14 (11.66)	114 (9.64)	
**Firewoo**	yes	163 (18.21)		101 (88.59)	10 (18.86)	38 (31.66)	312 (26.4)	261.478 (*p* < 0.001 **) df = 3
	No	732 (81.78)		13 (11.40)	43 (81.13)	82 (68.33)	870 (73.60)	
**Coal**	yes	329 (36.76)		16 (16.33)	1 (1.88)	22 (18.33)	368 (31.13)	59.299 (*p* < 0.001 **) df = 3
	No	566 (63.24)		98 (85.96)	52 (98.11)	98 (81.66)	814 (68.86)	

Data are presented as frequency (percentage). F = Female, M = Male, MD = Motorbike drivers, UW = Outdoor Urban Workers, FSS = Fuel Station Sellers, WCP = Workers in Closed Places, SD = Standard deviation, df = degree of freedom, Pearson’s chi-square test was used to compare proportions. ** = *p*-value less than 0.001 was considered significant; bold values = total values.

**Table 2 ijerph-18-00665-t002:** Exposure to air pollution in the study population.

Duration of Exposition		MD *n* = 895	UW *n* = 114	FSS *n* = 53	Total *n* =1062	χ2 (*p*-Value)
**Duration of the exposure to air (years) *n* (%)**	≤6	45 (05.00)	1 (0.80)	5 (8.90)	51 (4.80)	9.36
	(7–14)	698 (78.00)	97 (85.10)	45 (80.40)	840 (79.09)	(*p* = 0.0527 *)
	≥14	152 (17.00)	16 (14.10)	3 (5.40)	171 (16.10)	df = 4
**Daily exposure (time per hour) in (%)**	≤7	588 (65.70)	82 (71.90)	32 (57.10)	702 (66.10)	4.96
	(8–20)	292 (32.60)	29 (25.50)	21 (37.50)	342 (32.20)	(*p* = 0.2935)
	≥21	15 (01.70)	3 (02.60)	0 (0.00)	18 (1.69)	df = 4

Data are presented as frequency (percentage). F = Female, M = Male, MD = Motorbike drivers, UW = Outdoor Urban Workers, FSS = Fuel Station Sellers, df = degree of freedom, Pearson’s chi-square test was used to compare proportions. * = Correlation was significative for *p*-value less than 0.05.

**Table 3 ijerph-18-00665-t003:** Distribution of symptoms and diseases (*n* and %) in the studied populations.

Symptoms of Upper Airways *n* (%)	MD *n* = 895	UW *n* =114	FSS *n*= 53	WCP *n* =120	χ2 (*p*-Value)
Sinusitis	62 (3.48)	2 (2.56)	2 (3.22)	2 (6.66)	46.28
Running nose	470 (26.43)	16 (20.51)	9 (14.51)	2 (6.66)	
Cold	754 (42.4)	30 (38.46)	30 (48.38)	20 (66.66)	
Current fever	230 (12.93)	26 (33.33)	14 (22.58)	4 (13.33)	(*p* < 0.001)
Sore throat	262 (14.73)	4 (5.12)	7 (11.29)	2 (6.66)	
**Total**	**1778 (100)**	**78 (100)**	**62 (100)**	**30 (100)**	
**Symptoms of Lower Airways *n* (%)**	**MD *n*= 895**	**UW *n*= 114**	**FSS *n* = 53**	**WCP *n* = 120**	**χ2 (*p*-Value)**
Dry cough	679 (45.32)	23 (42.59)	33 (53.22)	8 (40)	163.85
Wheezing breath	136 (9.07)	27 (50)	3 (4.83)	12 (60)	
Chest discomfort	422 (28.17)	2 (3.7)	8 (12.9)	0 (0)	(*p*-value = 0.00001 **)
Breathlessness	261 (17.42)	2 (3.7)	18(29.032)	0 (0)	
**Total**	**1498 (100)**	**54 (100)**	**62 (100)**	**20 (100)**	
**Symptoms Related to Discomforts**	**MD *n*= 895**	**UW *n*= 114**	**FSS *n* = 53**	**WCP *n* = 120**	**χ2 test**
Headache	711 (24.82)	67 (29.51)	41 (41)	12 (26.66)	
Dizziness	75 (2.61)	13 (5.72)	7 (7)	0 (0)	
eyes irritation	632 (22.06)	31 (13.65)	10 (10)	0 (0)	(*p* < 0.001 **)
Conjunctivitis	282 (9.84)	31 (13.65)	0 (0)	0 (0)	
watering	516 (18.01)	25 (11.01)	7 (7)	0 (0)	
general tiredness	566 (19.76)	33 (14.53)	18 (18)	4 (8.88)	
Nausea	82 (2.86)	27 (11.89)	17 (17)	29 (64.44)	
**Total**	**2864 (100)**	**227 (100)**	**100 (100)**	**45 (100)**	

MD = Motorbike drivers, UW = Outdoor Urban Workers, FSS = Fuel Station Sellers, WCP = Workers in Closed Places, Pearson’s chi-square test was used to compare proportions. ** = Correlation is significative for *p*-value less than 0.001. Bold values = total values.

**Table 4 ijerph-18-00665-t004:** Distribution symptoms among motorbike drivers and group control selected for nested study.

Different Symptoms Related to Bad Air Quality		MD *n* = 126	Group Control *n* = 65	χ^2^ (*p*-Value)
**Symptoms of upper respiratory airways *n* (%)**				
	Sinusitis	11 (4.54)	6 (9.23)	0.013 (*p* = 0.908) 5.949 (*p* = 0.015 *) 25.090 (*p* < 0.001 **) 0.015 (*p* = 0.902) 9.752 (*p* = 0.002 *)
	Running nose	62 (25.61)	20 (25.00)	
	Cold	108 (44.62)	34 (42.50)	
	Current fever	30 (12.39)	16 (20.00)	
	Sore throat	31 (12.80)	4 (5.00)	
	**Total**	**242**	**80**	
**Symptoms of lower respiratory airways *n* (%)**				
	Dry cough	101 (39.45)	30 (36.14)	23.013 (*p* < 0.001 **)
	Wheezing breath	28 (10.93)	17 (20.48)	0.368 (*p* = 0.544)
	Chest discomfort	63 (24.60)	21 (25.30)	5.448 (*p* = 0.020 *)
	Breathless	64 (25)	15 (18.07)	13.582 (*p* < 0.001 **)
	**Total**	**256**	**83**	
**Systemic symptoms *n* (%)**				
	Headache	103 (26.68)	36 (29.03)	15.040 (*p* < 0.001 **)
	Dizziness	6 (1.55)	7 (5.64)	2.440 (*p* = 0.118)
	eyes irritation	88 (22.79)	26 (20.96)	15.869 (*p* < 0.001 **)
	Conjunctivitis	48 (12.43)	14 (11.29)	5.362 (*p* = 0.021 *)
	Watering	60 (15.54)	10 (8.06)	19.190 (*p* < 0.001 **)
	General tiredness	74 (19.17)	22 (17.74)	10.621 (*p* < 0.001 **)
	Nausea	7 (1.81)	9 (7.25)	3.840 (*p* = 0.051)
	**Total**	**386**	**124**	

Data are presented as frequency (percentage) and mean (±standard deviation). MD = Motorbike drivers. Pearson’s chi-square test was used to compare proportions. * = Correlation was significative for *p*-value less than 0.05. ** = Correlation is significative for *p*-value less than 0.001. Bolded values = total values.

**Table 5 ijerph-18-00665-t005:** Anthropometric parameters and haematological parameters and C-reactive protein in motorbikes.

Variable	MBD (Mean ± SD) *n* =126	Others (Mean ± SD) *n* = 65	T-Student *p*-Value
Systolic arterial tension	138.49 ± 16.56	126.55 ± 3.12	0.0001
Diastolic arterial tension.	87.72 ± 14.457	77.31 ± 4.586	0.0001
Biological parameters		Reference	
C-reactive protein (mg/dl)	1.0438 ± 1.83315	0.8012 ± 0.94985 (≤0.6 mg/dL)	0.230
Red blood cell (×10^12^ Cell/L)	4.9794 ± 0.53334	5.0932 ± 0.8557 (4.5–5.9)	0.330
Haemoglobin (g/dL)	14.6990 ± 1.55235	14.4015 ± 1.46815 (13.0–17.0)	0.196
Hematocrit (%)	43.9982 ± 4.75811	41.3615 ± 4.73454 (39–54)	0.0001
Neutrophils (%)	34.5017 ± 11.55805	47.3740 ± 12.550 (42.0–85.0)	0.0001
Monocytes (%)	15.5553 ± 6.24152	4.9931 ± 2.54159 (0.0–10)	0.0001
Lymphocytes (%)	50.0995 ± 10.21567 (20–40)	47.8129 ± 11.656 (20–40)	0.183
Platelets (× 10^9^ Cell./L)	234.8175 ± 39.54833 (150–500)	250.5846 ± 78.1846 (150–500)	0.130

Data are presented as mean ± standard deviation (SD). Student’s *t*-test was used to compare proportions between different MBD and controls groups. *p*-value less than 0.05 and 0.001 were considered significant.

**Table 6 ijerph-18-00665-t006:** Influence of exposure factors on the changes of some blood markers.

Exposure Factors		C-Reactive Protein	Neutrophils	Monocytes	Lymphocyte	Number of Subjects
Place of residence	R Pearson	0.006	−0.120	−0.194 *	0.255 *	191
	*p*-value	0.948	0.180	0.030	0.004	
Site of activity	R Pearson	0.069	0.21^*^	−0.194 *	0.109	191
	*p*-value	0.444	0.019	0.030	0.226	
Duration of activity (per year)	R Pearson	0.311 **	0.000	0.073	−0.060	191
	*p*-value	0.00001	0.998	0.414	0.504	
Daily duration (per hour)	R Pearson	0.042	0.199 *	0.091	−0.276 *	191
	*p*-value	0.638	0.026	0.312	0.002	

Bivariate correlation using Pearson’s test was used for assessing associations between sites and time of exposure, and each category of biological parameters. * Correlation was significative for *p*-value less than 0.05. ** Correlation is significative for *p*-value less than 0.001. R = Regression Coefficient.

**Table 7 ijerph-18-00665-t007:** Influence of exposure factors on the changes of some blood markers after adjustment for for age, level of education, body mass index (BMI), smoking and use of domestic gas, firewood and coal.

Exposure Factors	C-Reactive Protein R (*p*-Value)	Red Blood Cell R (*p*-Value)	Hemoglobin R (*p*-Value)	Neutrophils R (*p*-Value)	Monocytes R (*p*-Value)	Lymphocytes R (*p*-Value)	Platelettes R (*p*-Value)	Number of Subjects
**Mean Age**	−0.001 (0.965)	0.011 (0.249)	−0.003 (0.906)	−0.116 (0.532)	−0.223 (0.028)	0.339 (0.023 *)	0.449 (0.516)	191
**BMI**	−0.024 (0.483)	−0.002 (0.880)	0.013 (0.655)	−0.497 (0.051)	0.195 (0.159)	0.271 (0.180)	0.800 (−0.209)	191
**Categorized Age**	2.145 (0.099)	2.252 (0.087)	3.108 (0.030*)	1.166 (0.326)	1.345 (0.264)	0.213 (0.887)	0.044 (2.797)	191
**Level of education n (%)**	0.063 (2.507)	1.345 (0.264)	1.200 (0.314)	0.157 (0.925)	0.440 (0.725)	0.431 (0.731)	0.371 (1.057)	191
**Place of residence**	0.011 (3.928)	8.044 (0.0001 **)	4.763 (0.004 *)	2.104 (0.104)	6.444 (0.0001 **)	1.581 (0.198)	0.300 (1.237)	191
**Site of activity**	0.19 (0.663)	0.015 (0.903)	0.005 (0.946)	5.299 (0.023 *)	3.293 (0.072)	1.743 (0.190)	0.927 (0.008 *)	191
**Duration of activity (per year)**	0.0001 (10.678)	0.615 (0.543)	1.104 (0.335)	0.766 (0.468)	0.328 (0.721)	1.846 (0.163)	0.985 (0.015 *)	191
**Daily duration (per hour)**	0.255 (0.615)	0.148 (0.701)	0.323 (0.571)	(0.209) 1.598	0.106 (0.745)	2.823 (0.096)	0.738 (0.113)	191
**Consumption of tabac**	6.971 (0.001)	2.064 (0.132)	0.808 (0.449)	1.738 (0.181)	0.108 (0.897)	1.429 (0.244)	0.921 (0.082)	191
**Domestic gas**	0.665 (0.417)	0.265 (0.608)	0.808 (0.449)	0.506 (0.478)	2.796 (0.097)	0.059 (0.808)	0.252 (0.617)	191
**firewood**	1.335 (0.251)	0.732 (0.394)	0.525 (0.470)	0.190 (0.664)	0.003 (0.956)	0.044 (0.834)	0.404 (0.526)	191
**coald**	0.510 (0.477)	0.143 (0.706)	0.244 (0.622)	0.026 (0.873)	0.359 (0.550)	0.898 (0.345)	0.085 (0.771)	191

Linear regression and multinominal analysis were used for assessing associations between sites, time of exposure (in year and hour), after adjustment for age, level of education, Body Mass Index (BMI), consumption of tabac and use of domestic gas, firewood and coald, for each category of biological parameters. * Correlation was significative for *p*-value less than 0.05. ** Correlation is significative for *p*-value less than 0.001. R = regression coefficient.

**Table 8 ijerph-18-00665-t008:** Correlation between symptoms of upper airways and changes of haematological markers.

Haematological Markers		Sinusitis	Running Nose	Cold	Current Fever	Sore Throat	Number of Subjects
c-reactive protein	R Pearson	0.332 **	0.069	0.097	0.047	0.010	191
	*p* value	0.0001	0.446	0.281	0.598	0.908	
Red blood cell (×10^12^/L)	R Pearson	0.137	0.0001	0.181 *	0.058	0.046	191
	*p* value	0.126	0.996	0.043	0.521	0.609	
Haemoglobin (g/dL)	R Pearson	0.139	0.065	0.046	0.152	0.133	191
	*p* value	0.119	0.467	0.606	0.090	0.138	
Hematocrit (%)	R Pearson	0.091	0.005	0.0001	0.100	0.092	191
	*p* value	0.310	0.955	0.999	0.264	0.307	
Neutrophils (%)	R Pearson	−0.260 *	0.070	0.121	−0.093	0.024	191
	*p* value	0.003	0.435	0.178	0.300	0.792	
Monocytes (%)	R Pearson	0.160	−0.038	−0.315 **	0.047	−0.054	191
	*p* value	0.073	0.674	0.0001	0.600	0.550	
Lymphocytes (%)	R Pearson	0.213^*^	−0.041	0.110	0.098	−0.040	191
	*p* value	0.017	0.648	0.221	0.276	0.659	
Platelets (×10^9^ Cell./L)	R Pearson	−0.182 *	0.040	0.051	0.004	−0.178 *	191
	*p* value	0.041	0.656	0.574	0.969	0.046	

* Bivariate correlation using Pearson’s test was used for assessing associations between symptoms and and each category of biological parameters. * = Correlation was significative for *p*-value less than 0.05. ** Correlation is significative for *p*-value less than 0.001. R = regression coefficient.

**Table 9 ijerph-18-00665-t009:** Correlation between symptoms of lower airways and changes of haematological markers.

Haematological Markers		Dry Cough	Wheezing Breath	Chest Discomfort	Breathlessness	Number of Subjects
C-reactive protein	R Pearson	−0.183 *	0.219 *	−0.004	−0.106	191
	*p*-value	0.041	0.014	0.963	0.235	
Red blood cell (×10^12^/L)	R Pearson	−0.058	0.187^*^	−0.072	0.001	191
	*p* value	0.517	0.036	0.422	0.993	
Hemoglobin (g/dL)	R Pearson	0.014	0.208^*^	−0.039	−0.037	191
	*p*-value	0.878	0.019	0.666	0.684	
Hematocrit (%)	R Pearson	−0.019	0.145	−0.013	−0.026	191
	*p*-value	0.833	0.105	0.888	0.777	
Neutrophils (%)	R Pearson	0.156	−0.097	−0.107	−0.085	191
	*p*-value	0.081	0.282	0.234	0.343	
Monocytes (%)	R Pearson	−0.164	−0.113	0.165	0.003	191
	*p*-value	0.066	0.207	0.065	0.970	
Lymphocytes (%)	R Pearson	−0.114	0.159	0.014	0.122	191
	*p*-value	0.205	0.076	0.878	0.173	
Platelets (×10^9^ Cell./L)	R Pearson	0.082	0.103	0.010	0.001	
	*p*-value	0.364	0.252	0.913	0.987	

Bivariate correlation using Pearson’s test was used for assessing associations between symptoms and each category of biological parameters. * Correlation is significative for *p*-value less than 0.05. R = regression coefficient.

**Table 10 ijerph-18-00665-t010:** Correlation between symptoms related to discomforts and changes of haematological markers.

Haematological Markers		Headache	Dizziness	Eyes Irritation	Conjunctivitis	Watering	General Tiredness	Nausea	Number of Subjects
c-reactiv protein	R Pearson	0.001	0.146	0.017	−0.097	−0.025	0.050	0.067	191
	*p*-value	0.993	0.103	0.849	0.282	0.778	0.581	0.458	
Red blood cell (×10^12^/L)	R Pearson	0.025	0.040	0.011	0.125	0.016	0.064	0.057	191
	*p* value	0.784	0.660	0.902	0.164	0.857	0.476	0.526	
Hemoglobin (g/dL)	R Pearson	0.023	0.043	0.120	−0.023	−0.004	0.051	0.101	191
	*p*-value	0.794	0.636	0.182	0.796	0.966	0.571	0.259	
Hematocrit (%)	R Pearson	0.026	0.041	0.074	0.022	−0.081	0.055	0.066	191
	*p*-value	0.773	0.647	0.409	0.805	0.370	0.538	0.461	
Neutrophils (%)	R Pearson	−0.010	−0.217 *	−0.073	0.103	0.118	−0.164	−0.007	191
	*p*-value	0.910	0.015	0.414	0.251	0.188	0.066	0.942	
Monocytes (%)	R Pearson	−0.020	0.073	−0.041	−0.122	−0.172	0.120	0.052	191
	*p*-value	0.826	0.417	0.652	0.174	0.054	0.181	0.563	
Lymphocytes (%)	R Pearson	−0.006	0.170	0.095	−0.060	−0.095	0.119	−0.030	191
	*p*-value	0.944	0.057	0.290	0.504	0.291	0.184	0.740	
Platelets (×10^9^ Cell./L)	R Pearson	−0.035	−0.014	0.055	0.039	−0.050	−0.200 *	−0.098	191
	*p*-value	0.701	0.875	0.541	0.662	0.578	0.025	0.274	

* Bivariate correlation using Pearson’s test was used for assessing associations between symptoms and and each category of biological parameters. R = regression coefficient.

## Data Availability

The data presented in this study are available on request from the corresponding author. The data are not publicly available due to privacy of datas, but will be available at the moment.

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
