# Peer review of "Clinical Manifestations and Changes of Haematological Markers among Active People Living in Polluted City: The Case of Douala, Cameroon"

_ijerph, 2021, doi:10.3390/ijerph18020665_

Round 1
Reviewer 1 Report
This is an interesting study investigating possible effects on air pollution and heamatological markers.
I have only two comments to the study presentation, would it be possible to look at assosiations between the health record or questionnaire from each person and the heamatological findings, to see if there are true associations in these markers?
The second is to better describe the 3 numbers in table 2, why are there 3 numbers in each column?
Minor comments is to check for small typos in the text.
Author Response
Response to Reviewer 1 Comments
Point 1: I have only two comments to the study presentation, would it be possible to look at assosiations between the health record or questionnaire from each person and the heamatological findings, to see if there are true associations in these markers?
Response 1: In the amended version of our paper we have looked at the association between health status as reported by the questionnaire and the haematological finding in each category of persons (motorbike drivers and control group). We have added these tables (line 79 to line 103) , and comments
Point 2: The second is to better describe the 3 numbers in table 2, why are there 3 numbers in each column?
Response 2: The 3 numbers in table 2 represent the numbers of exposed people according to either the time of daily exposure in hour or in a year of activity. These time were categorized in three parts
Reviewer 2 Report
1) I appreciate the opportunity to review this paper titled, 'Clinical Manifestations and Changes of Heamatological Markers Related to Urban Air Pollution Among Active People in Douala, Cameroon'. Firstly, I do not understand why the title is all in Capital Letters. And secondly, this title is totally misleading because the paper does not present any air pollution data.
2) The authors have anecdotally tried to establish a linkage between these various blood markers and air pollution without using any air pollution data. The authors might want to change the title of the manuscript to avoid confusion.
3) I am also wondering how did the authors come to the conclusion that air pollution is the main culprit agent here. What about other point and area sources of air pollution? Cameroon is a low-middle income country and it is quite possible that household pollution from biomass burning, industrial pollution, pollution from agricultural emissions may be the many confounding factors resulting in these changes in blood markers. How about nutrition, exercise, genetics, and other probable causes?
4) The manuscript also requires a major English revision as the sentence structure, grammar, and syntax are all erroneous. It seems the paper is written primarily by researchers whose official working language is French. Please get the paper proof checked by native English speakers - either British or American.
5) The discussion and conclusion part has huge segments discussing the PM and other pollutants and their general effects on human health. At a lot many places, the citations/references are missing. The discussion should focus more on the findings of the paper and the importance and significance of this research work. Rather, what I see is a lot of padding in the manuscript just to increase the word count.
6) Table 5 talks about Pearson's Correlations and Multivariate Analysis. But the authors have just provided the Pearson's R. Where are the findings from the multivariate analysis? I personally feel that the paper lacks robust statistical analyses !
7) Finally, the affiliation of the 4th and the 5th author are missing. Their emails and as well as their institutional information is not presented. Please rectify this.
8) In sum, this paper needs substantial revisions. It is only after these revisions are undertaken can a decision be made regarding the acceptance of this manuscript.
Author Response
Response to Reviewer 2 Comments
Point 1: I appreciate the opportunity to review this paper titled, 'Clinical Manifestations and Changes of Heamatological Markers Related to Urban Air Pollution Among Active People in Douala, Cameroon'. Firstly, I do not understand why the title is all in Capital Letters. And secondly, this title is totally misleading because the paper does not present any air pollution data.
Response 1: The title has been changed, as suggested by the reviewer.
Point 2: The authors have anecdotally tried to establish a linkage between these various blood markers and air pollution without using any air pollution data. The authors might want to change the title of the manuscript to avoid confusion
Response 2: The title has been changed, as suggested by the reviewer.
Point 3: I am also wondering how did the authors come to the conclusion that air pollution is the main culprit agent here. What about other point and area sources of air pollution? Cameroon is a low-middle income country and it is quite possible that household pollution from biomass burning, industrial pollution, pollution from agricultural emissions may be the many confounding factors resulting in these changes in blood markers. How about nutrition, exercise, genetics, and other probable causes?
Response 3: We agree with the reviewer. In the discussion, we quote all the air pollution sources. They are common to most of the study participants, with the exception of urban air pollution to which motobikers were more exposed in terms of proximity and length of exposure.
Point 4: The manuscript also requires a major English revision as the sentence structure, grammar, and syntax are all erroneous. It seems the paper is written primarily by researchers whose official working language is French. Please get the paper proof checked by native English speakers - either British or American.
Response 4: We thank the reviewer for having noticed it. Cameroon is a bilingual speaking country. We have the paper checked by an english speaking colleague.
Point 5: The discussion and conclusion part has huge segments discussing the PM and other pollutants and their general effects on human health. At a lot many places, the citations/references are missing. The discussion should focus more on the findings of the paper and the importance and significance of this research work. Rather, what I see is a lot of padding in the manuscript just to increase the word count.
Response 5: We have reduced the discussion by targeting exclusively the findings of the paper.
Point 6: Table 5 talks about Pearson's Correlations and Multivariate Analysis. But the authors have just provided the Pearson's R. Where are the findings from the multivariate analysis? I personally feel that the paper lacks robust statistical analyses !
Response 6: This is exact. Results about the multivariate analysis are presented in the text in the amended version of our paper.
Point 7: Finally, the affiliation of the 4th and the 5th author are missing. Their emails and as well as their institutional information is not presented. Please rectify this.
Response 7: Affiliations have been added.
Point 8: In sum, this paper needs substantial revisions. It is only after these revisions are undertaken can a decision be made regarding the acceptance of this manuscript.
Response 8: We have reviewed our paper.
Round 2
Reviewer 2 Report
I appreciate the opportunity to review the revised version of this manuscript from Cameroon. The authors have incorporated all my suggestions as requested. I would, therefore, recommend the publication of this manuscript. I do not have any further comments at this stage.